# Re³Dial: Retrieve, Reorganize and Rescale Conversations for Long-Turn Open-Domain Dialogue Pre-training

**Jiaxin Wen[1,2], Hao Zhou[3], Jian Guan[1,2], Jie Zhou[3], Minlie Huang[1,2,†]**

[1]The CoAI group, Tsinghua University, Beijing, China

[2]Department of Computer Science and Technology, Tsinghua University, Beijing, China

[3]Pattern Recognition Center, WeChat AI, Tencent Inc., China

wenjx22@mails.tsinghua.edu.cn, aihuang@tsinghua.edu.cn

## Abstract

Pre-training on large-scale open-domain dialogue data can substantially improve the performance of dialogue models. However, the pre-trained dialogue model's ability to utilize long-range context is limited due to the scarcity of long-turn dialogue sessions. Most dialogues in existing pre-training corpora contain fewer than three turns of dialogue. To alleviate this issue, we propose the **Re**trieve, **Re**organize and **Re**scale framework (Re³Dial), which can automatically construct billion-scale long-turn dialogues by reorganizing existing short-turn ones. Given a short-turn session, Re³Dial first employs a session retriever to retrieve coherent consecutive sessions. To this end, we train the retriever to capture semantic and discourse relations within multi-turn dialogues through contrastive training. Next, Re³Dial samples a session from retrieved results following a diversity sampling strategy, which is designed to penalize repetitive or generic sessions. A longer session is then derived by concatenating the original session and the sampled session. By repeating the above process, Re³Dial can yield a coherent long-turn dialogue. Extensive experiments on multiple multi-turn dialogue benchmarks demonstrate that Re³Dial significantly improves the dialogue model's ability to utilize long-range context and thus generate more sensible and informative responses. Finally, we build a toolkit for efficiently rescaling conversations with Re³Dial, which enables us to construct a corpus containing 1B Chinese dialogue sessions with 11.3 turns on average (5× longer than the original corpus). Our retriever model, code, and data is publicly available at https://github.com/thu-coai/Re3Dial.

## 1 Introduction

Building intelligent open-domain dialogue systems that can generate coherent and engaging multi-turn dialogues with humans has been one of the long-

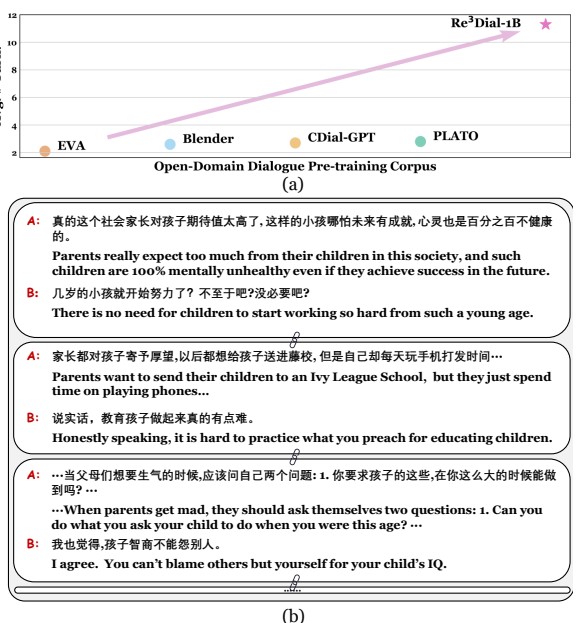

Figure 1: (a) Statistics of several open-domain dialogue pre-training corpora, none of which has more than 3 turns on average. Taking EVA as an example, Re³Dial can construct a new corpus with 1B sessions and 11.3 turns on average, which is 5× longer than that of the EVA corpus. (b) An excerpt of the automatically constructed long-turn dialogue by Re³Dial. More detailed examples are presented in Appendix G.

standing goals in AI. Recently, a variety of large-scale open-domain pre-trained dialogue models have dramatically promoted this progress (Roller et al., 2020; Zhou et al., 2021; Shuster et al., 2022b). And a critical ingredient to the success of these models is the pre-training dialogue corpus. However, while existing dialogue pre-training corpus collects millions to billions of dialogues from public social media, e.g., Reddit for English (Roller et al., 2020) and Weibo for Chinese (Zhou et al., 2021), long-turn dialogues are highly scarce. More specifically, based on the publicly reported data statistics shown in Figure 1(a), *most dialogues in existing pre-training corpora only have less than three turns*. The lack of large-scale long-turn di-

---

† Corresponding author

alogue data restricts dialogue models from deriving more advanced abilities to utilize long-range context for modeling multi-turn dialogues during pre-training (Xu et al., 2021, 2022b).

In this paper, we focus on answering the following research question:

> *Can we automatically build a billion-scale long-turn dialogue corpus by reorganizing existing short-turn dialogues?*

Our basic idea is to construct a long-turn dialogue via recursively retrieving and selecting one consecutive session from the existing dialogue corpus. Despite the simplicity of this idea, we still face several challenges to make the constructed corpus effective in enhancing long-turn dialogue pre-training. **First**, the selected session should be coherent with the query session. Otherwise, it will introduce noisy utterances without long-range dependency or break the conversation flow (Liu et al., 2021), which may impact the performance of dialogue models. **Second**, our in-depth analysis reveals that the retrieved sessions tend to be biased to be relevant but semantically repetitive with the query or overly generic (e.g., "*A: Haha, it's so cute. B: Haha! LMAO.*") due to both the data bias in the dialogue corpus (Zhou et al., 2021; Lee et al., 2021; Li et al., 2015; Liu et al., 2018) and the model bias of the retriever (Thakur et al., 2021). These biases significantly lower the diversity and informativeness of the reorganized long-turn dialogues.

To tackle the above challenges, we propose the **Re**trieve, **Reorganize** and **Re**scale framework (Re³Dial), which employs an *Unsupervised Dense Session Retriever* (UDSR) to retrieve coherent short-turn dialogues and reorganize them into a long-turn one. We train UDSR through contrastive learning by taking consecutive dialogue segments from the same dialogue as positive pairs and those from different dialogues as negative pairs. To avoid overly retrieving semantically repetitive or generic sessions, we propose a diversity sampling strategy, effectively improving the diversity and informativeness of the reorganized long-turn dialogues. Figure 1(b) shows an example of the automatically constructed long-turn dialogue using Re³Dial.

We verify the effectiveness of Re³Dial on three Chinese multi-turn open-domain dialogue benchmarks. Extensive experiments demonstrate that Re³Dial consistently and significantly enhances the dialogue model's ability to utilize long-range context, leading to more sensible and informative

responses in multi-turn dialogue. Finally, we develop a toolkit for efficiently rescaling conversations with Re³Dial, which enables us to construct a corpus containing 1B Chinese dialogue sessions with 11.3 turns on average ($5\times$ longer than that of the original EVA corpus). We will make our retriever model, toolkit, and data public. We believe our work provides new opportunities in long-turn dialogue pre-training to the research community.

Our contributions can be summarized as follows:

- We introduce Re³Dial, which presents a novel perspective to alleviate the scarcity of long-turn conversations by automatically building a billion-scale long-turn dialogue corpus via concatenating existing short-turn dialogue.

- We propose to train a dense session retriever on massive unlabeled plain dialogue data with contrastive learning to capture the global semantic and discourse relations within multi-turn dialogues. We also propose the diversity sampling strategy to improve the diversity and informativeness of the automatically constructed corpus.

- Experiments on three Chinese multi-turn dialogue benchmarks demonstrate that Re³Dial can enhance the model's ability to model long-range context, thereby leading to consistent and significant improvements in different pre-training settings.

- We release Re³Dial-1B, which contains 1B Chinese dialogue with 11.3 turns on average ($5\times$ longer than that of the original EVA corpus).

## 2 Related Work

### 2.1 Large-Scale Open-Domain Dialogue Pre-training

In the past few years, large-scale pre-training has greatly promoted the progress of the NLP community (Brown et al., 2020). Recently, large-scale pre-training has also become the mainstream approach to building open-domain dialogue models, both in English (Zhang et al., 2019; Roller et al., 2020; Thoppilan et al., 2022) and Chinese (Bao et al., 2020; Zhou et al., 2021; Gu et al., 2022; Wen et al., 2022). Through pre-training on massive dialogue data crawled from public social media, these models exhibit strong conversational ability, significantly outperforming traditional non-pre-trained dialogue models. However, the scarcity of long-turn dialogues in the pre-training corpus hinders

these models from deriving a better ability to utilize long-range context for modeling multi-turn dialogues during pre-training. To alleviate this issue, we study how to automatically and efficiently build a large-scale long-turn dialogue corpus based on the existing short-turn dialogue corpus.

## 2.2 Retrieval-Augmented Language Model

Extending neural language models with a retrieval system has been widely studied in various NLP tasks, such as language modeling (Khandelwal et al., 2019), story generation (Zhang et al., 2022), and open-domain QA (Lewis et al., 2020; Izacard and Grave, 2020). The integration of retrieval techniques in open-domain dialogue systems also has a long history. Retrieval-based dialogue systems directly return a response via retrieving from a large dialogue corpus (Ji et al., 2014; Zhou et al., 2018). Moreover, recent works investigate how to generate more accurate responses via retrieving from external knowledge sources (Komeili et al., 2021; Shuster et al., 2022a). While we also leverage a retrieval system in open-domain dialogue, our work is significantly distinguished from these mainly in that: (1) We aim to enhance the model's ability to utilize long-range context for modeling multi-turn dialogue rather than focusing on improving factuality or directly responding. (2) We leverage the retrieval system only for constructing a long-turn dialogue corpus, which is disentangled from the training and inference stages of dialogue models. Consequently, our approach does not introduce additional training costs or inference latency.

## 3 Methodology

An overview of Re³Dial is shown in Figure 2. Let $D = \{S_i\}$ be the original dialogue pre-training corpus. For each session $S_i$, Re³Dial constructs long-turn dialogues automatically in four steps: **(1)** Initialize the constructed session $S_{out}$ with $S_i$: $S_{out} = S_i$. **(2)** Use UDSR to retrieve top-$K$ coherent sessions from $D$: UDSR$(S_i^1) = \{S_{c_i}^1, S_{c_i}^2, \cdots, S_{c_i}^K\}$. **(3)** Use diversity sampling to sample a consecutive session $\hat{S_{c_i}}$. **(4)** Update $S_{out}$ and obtain a longer session by $S_{out} = S_{out} \oplus \hat{S_{c_i}}$. Let $\hat{S_{ci}}$ be $S_i$. Go to step 2 until $S_{out}$ has been updated for $L$ times. $L$ is a hyperparameter to control the number of turns of the constructed dialogue.

---

[1]We use the last session $S_i$ instead of the full context $S_{out}$ for retrieval since the representation of $S_{out}$ needs to be dynamically computed during the iterative retrieval process, leading to a significant increase in time cost. More analysis

## 3.1 Retrieve

**Task Definition** We define the dialogue session retrieval task as follows. Given a dialogue session with $|Q|$ utterances $S_q = \{u_q^1, \cdots, u_q^{|Q|}\}$, our goal is to choose a $|C|$-turn dialogue session $S_c = \{u_c^1, \cdots, u_c^{|C|}\}$ from a dialogue corpus that should be coherent with $S_q$. Consequently, $S_q \oplus S_c = \{u_q^1, \cdots, u_q^{|Q|}, u_c^1, \cdots, u_c^{|C|}\}$ can make a natural dialogue session of $|Q| + |C|$ turns.

We observe that previous retrievers either rely solely on local term matching and fail to capture global semantic relations (e.g., BM25 (Robertson et al., 2009)) or struggle to capture discourse coherence within multi-turn dialogues (Liu et al., 2021) (e.g., Contriever (Izacard et al., 2022)). Therefore, these retrievers exhibit unsatisfactory performance in our dialogue session retrieval task. To remedy these problems, we train an Unsupervised Dense Session Retriever (UDSR). By using consecutive dialogue segments from the same dialogue as positive pairs and those from different dialogues as negative pairs, UDSR demonstrates superior capability in capturing global semantic relevance and discourse coherence within multi-turn dialogues.

**Model Structure** Given two dialogue sessions $S_q$ and $S_c$, we encode them using two encoder models, $E_q$ and $E_c$. The similarity score is defined as the dot product of their representations: $E_q(S_q)^{\mathrm{T}} E_c(S_c)$.

**Contrastive Training** We adopt contrastive training to train the dialogue session encoder $E_q$ and $E_c$. For each training instance $\{S_q, S_c^+, S_{c_1}^-, \cdots, S_{c_n}^-\}$, which contains one query session $S_q$, one coherent session $S_c^+$, and $n$ incoherent sessions $\{S_{c_i}^-\}_{i=1\ldots n}$, we optimize the contrastive loss $\mathcal{L}$ as follows:

$$\mathcal{L} = -log\frac{e^{\text{sim}(S_q,S_c^+)}}{e^{\text{sim}(S_q,S_c^+)} + \sum_{i=1}^n e^{sim(S_q,S_{c_i}^-)}}$$

**Positive and Negative Pairs** Large-scale positive and negative pairs are crucial for the effectiveness of contrastive learning (Izacard et al., 2022). However, considering that there is no available labeled data for this task, we propose to build positive and negative pairs from unlabeled plain dialogues, which are much easier to access. Let $\{S_{u_i}\}$ be an unlabeled dialogue corpus, where $S_{u_i} = \{S_{u_i}^1, \cdots, S_{u_i}^{K_i}\}$ is a $K_i$-turn

---

of the retrieval performance with varying context lengths is illustrated in Appendix E.4.

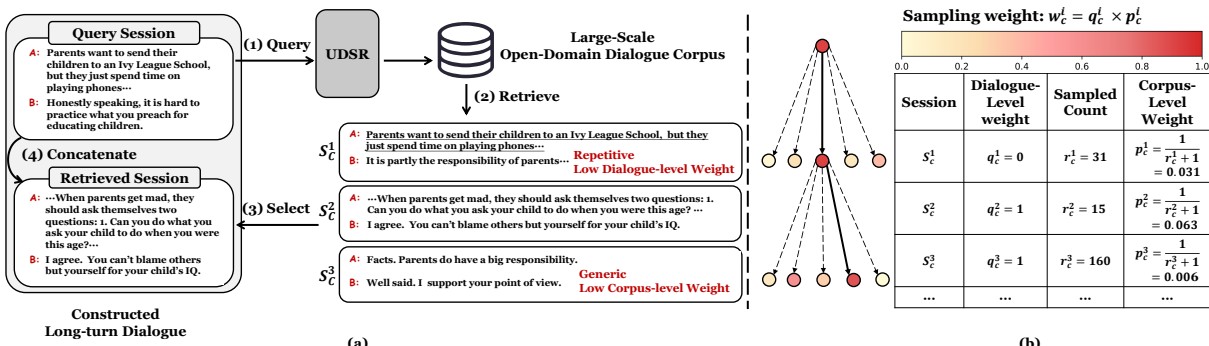

Figure 2: Overview of Re³Dial. (a) Constructing multi-turn dialogues via recursively leveraging UDSR to retrieve consecutive sessions from a large-scale open-domain dialogue corpus. (b) The proposed diversity sampling strategy assigns each retrieved session a sampling weight, which is a combination of dialogue-level weight and corpus-level weight, aiming to avoid overly repetitive or generic retrieved sessions.

session. We first divide each $S_{u_i}$ into two consecutive segments $S_{q_i} = \{S_{u_i}^1, \cdots, S_{u_i}^{M_i}\}$ and $S_{c_i} = \{S_{u_i}^{M_i+1}, \cdots, S_{u_i}^{K_i}\}$, where $M_i$ is randomly chosen from $[2, K_i - 2]$. We then obtain a positive pair $(S_{q_i}, S_{c_j})$ if $i == j$. Moreover, we consider two kinds of negatives: (1) Easy Negatives: given $S_{q_i}$, we randomly select a consecutive session $S_{c_j}$ where $i \neq j$. (2) Hard Negatives: given $S_{q_i}$, we leverage the top-$K$ sessions retrieved by BM25 as $S_{c_i}$, thereby improving the model's ability to differentiate those incoherent negatives that have lexical overlaps (Huang et al., 2020).

## 3.2 Reorganize

After building the retriever, we can then reorganize the existing short-turn corpus into a long-turn corpus by recursively retrieving and selecting the consecutive session $\hat{S}_{c_i}$.

In this section, we first provide an in-depth analysis to reveal that the selected session tends to be biased towards relevant but repetitive or plausible but generic. These biases can be attributed to both the dataset bias of the original corpus and the model bias of the retriever. As shown in Section 4.4, these biases lead to reduced diversity and informativeness of the constructed corpus, finally resulting in decreased model performance. To remedy these problems, we introduce the diversity sampling strategy at both dialogue-level and corpus-level. Formally, for each retrieved session $S_{c_i}^k$, we derive its sampling weight $w_{c_i}^k$ as follows:

$$w_{c_i}^k = q_{c_i}^k \times p_{c_i}^k$$

where $q_{c_i}^k$ is a binary dialogue-level weight, and $p_{c_i}^k$ is a numeric corpus-level weight. We then adopt weighted sampling to select $\hat{S}_{c_i}$.

**Dialgoue-Level Diversity Sampling** There are widely duplicate substrings in the original dialogue pre-training corpus because: (1) Large-scale open-domain dialogues are mainly collected from public social media (Roller et al., 2020; Zhou et al., 2021) as follows. Given one post $P$ and multiple comments $\{C_i\}$, we derive multiple sessions $\{(P, C_i)\}$ which share the same prefix $P$. (2) There are generally duplicate contents in web-crawled datasets. For example, Lee et al. (2021) find that web-crawled datasets contain between 3.04% (on C4) to 13.63% (on RealNews) duplicate substrings. Moreover, such dataset bias in the original dialogue pre-training corpus will be amplified due to the model bias of the retriever since a higher lexical overlap generally leads to a higher similarity score, whether in sparse or dense latent space.

Then, the cross-sample duplicates would become in-context duplicates in the concatenated dialogue, e.g., $(P, C_1, P, C_2)$. We conjecture that such in-context duplication could bias the model towards simply copying context for response generation. To remedy this problem, we introduce the dialogue-level weight $S_q$, where we set $q_{c_i}^k = 0$ (1 otherwise) if it meets any of the two requirements: (1) Any utterance in $S_{c_i}^k$ is exactly matched with $S_{out}$. (2) The longest common substring (LCS) between $S_{out}$ and $S_{c_i}^k$ contains more than $N$ words.

**Corpus-Level Diversity Sampling** A lot of generic or meaningless utterances exist in the original dialogue corpus, e.g., "*A: Haha, it's so cute. B: Haha! LMAO.*" (Li et al., 2015; Liu et al., 2018). Due to their high frequency and compatibility with various dialogue contexts, the retriever is prone to select these plausible but generic dialogues as

| Pre-training Setting | Pre-trained Model | Model Architecture | Model Size |
|---|---|---|---|
| Pre-training From Scratch | - | non-causal decoder | 6B |
| Further Pre-training on LM | GPT2-small | causal decoder | 100M |
| Further Pre-training on DM | ChatGLM | encoder-decoder | 6B |

Table 1: Backbone model information for each pre-training setting.

consecutive sessions. Consequently, these generic sessions are more frequently sampled than more contentful and specific sessions at the corpus-level during reorganizing, which will decrease the informativeness and diversity of the final corpus, aggravating the problem of generic replies generated by dialogue models. To remedy this problem, we introduce the corpus-level weight $p_{c_i}^k$ to penalize a session for being repeatedly sampled:

$$p_{c_i}^k = \frac{1}{r_{c_i}^k + 1}$$

where $r_{c_i}^k$ is the sampled times of $S_{c_i}^k$.

### 3.3 Rescale

We finally build a toolkit for efficiently rescaling dialogue corpus with Re³Dial. We speed up retrieving with FAISS (Johnson et al., 2019), which achieves 192 searching per second on a single V100 GPU. Furthermore, we support parallel searching over multi-GPUs, which can achieve 1,536 searching per second with 8 V100 GPUs, for example. We also leverage PyArrow[2] to speed up the processing of big data. In practice, we show the efficiency of our toolkit by constructing Re³Dial-1B in 4.7.

## 4 Experiment

### 4.1 Retriever

We train UDSR on a subset of the EVA pre-training corpus (Zhou et al., 2021), which contains 1,000,000/49,000/1,000 examples for the train/validation/test split. More details of data processing are provided in Appendix A.1. We adopt BERT-base (Devlin et al., 2018) as the encoder backbone. The parameters of $E_q$ and $E_c$ are not shared according to our preliminary experiments.

### 4.2 Dialogue Model

**Settings** We consider three general scenarios where Re³Dial can be utilized for long-turn dialogue pre-training: (1) **Pre-training From Scratch**, where we pre-train a dialogue model from scratch. (2) **Further Pre-training on LM**, where

we further pre-train an existing pre-trained general language model. (3) **Further Pre-training on DM**, where we further pre-train an existing pre-trained dialogue model. Table 1 shows the detailed backbone model information for each setting.

**Pre-training** We extract a subset of the EVA pre-training corpus as the original corpus, which contains 5 Million dialogue sessions. For Re³Dial, we set $L$=5, top-$K$=5, and the maximum LCS length $N$=10. The average number of turns in the original corpus is 2.2, while for the Re³Dial-constructed corpus, it significantly increases to 11.6. We set the maximum sequence length to 256. For pre-training from scratch, we set the batch size to 512 and the pre-training steps to 10K. For further pre-training, we set the batch size to 256 or 128 and the pre-training steps to 30K. We pre-train the model with the auto-regressive language modeling task. More training details are shown in Appendix B.

**Benchmarks** We conduct evaluations on three widely-adopted Chinese open-domain multi-turn dialogue benchmarks, including KdConv (Zhou et al., 2020), DuLeMon (Xu et al., 2022b), and NaturalConv (Wang et al., 2021), each has 16~20 turns on average. Data statistics are shown in Table 9.

**Metrics** We adopt the following automatic metrics for evaluation. **PPL_zero-shot** measures the perplexity on the test set without fine-tuning on the downstream training sets. **PPL** measures the perplexity on the test set after fine-tuning. **BLEU-N** measures the precision of the n-gram overlap between generated and ground-truth responses (Papineni et al., 2002) after fine-tuning. **ROUGE-L** measures the recall of the n-gram overlap between generated and ground-truth responses (Lin, 2004) after fine-tuning. **Distinct-N** measures the percentage of the unique n-grams over all the generated n-grams after fine-tuning (Li et al., 2015).

### 4.3 Main Results

#### 4.3.1 Automatic Evaluation

Table 2 shows the automatic evaluation results. In the zero-shot setting, Re³Dial consistently outperforms the original baseline by a large margin in PPL_zero-shot on three benchmarks across different pre-training settings[3]. For instance, in the last block, the Re³Dial-trained model achieves a PPL of

---

[2] https://github.com/apache/arrow

---

[3] We also present zero-shot experiment results on English benchmarks in Appendix D.

| Benchmark | Pre-training Data | PPL$_{\text{zero-shot}}$ | PPL | BLEU-1 | BLEU-2 | ROUGE-L | Distinct-2 |
|---|---|---|---|---|---|---|---|
| *Pre-training From Scratch* | | | | | | | |
| **DuLeMon** | Original | 106.70 | 61.56 | 20.74 | 9.20 | 17.75 | 7.87 |
| | Re$^3$Dial | **83.10** | **56.09** | **21.43** | **9.66** | **18.75** | **8.32** |
| **KdConv** | Original | 309.82 | 42.95 | 23.83 | 13.25 | 21.28 | 6.60 |
| | Re$^3$Dial | **199.73** | **34.67** | **24.64** | **14.36** | **22.95** | **7.35** |
| **NaturalConv** | Original | 164.00 | 62.80 | 20.86 | 9.76 | 22.94 | 7.05 |
| | Re$^3$Dial | **124.80** | **57.28** | **22.14** | **10.39** | **23.35** | **7.98** |
| *Further Pre-training on LM* | | | | | | | |
| **DuLeMon** | Original | 12.95 | 10.07 | 15.10 | 8.03 | 19.06 | 13.78 |
| | Re$^3$Dial | **11.94** | **9.94** | **15.54** | **8.26** | **19.29** | **14.00** |
| **KdConv** | Original | 12.13 | 5.94 | 22.16 | 14.71 | 27.20 | **9.27** |
| | Re$^3$Dial | **11.56** | **5.80** | **23.17** | **15.41** | **27.56** | 9.24 |
| **NaturalConv** | Original | 14.11 | 10.56 | 16.48 | 8.48 | 21.89 | 13.94 |
| | Re$^3$Dial | **13.26** | **10.26** | **17.38** | **9.04** | **21.91** | **14.54** |
| *Further Pre-training on DM* | | | | | | | |
| **DuLeMon** | Original | 48.79 | 29.77 | 16.65 | 7.38 | 14.80 | 21.66 |
| | Re$^3$Dial | **46.25** | **29.27** | **17.12** | **7.55** | **15.07** | **22.28** |
| **KdConv** | Original | 60.08 | 10.87 | 21.90 | 13.71 | 21.38 | 16.83 |
| | Re$^3$Dial | **48.58** | **10.15** | **22.79** | **14.45** | **22.09** | **16.93** |
| **NaturalConv** | Original | 69.92 | 30.52 | 17.90 | 8.75 | 18.32 | 23.45 |
| | Re$^3$Dial | **67.78** | **29.20** | **18.54** | **9.10** | **18.59** | **24.24** |

Table 2: Automatic evaluation results. The best performance is highlighted in **bold**. Note that perplexity is not comparable across different settings since the backbone model uses different vocabulary.

46.25 on DuLeMon, compared to the original baseline's performance of 48.79. This indicates a better ability in multi-turn dialogue modeling. Moreover, beyond benefiting zero-shot performance, Re$^3$Dial can also significantly improve the model's performance after fine-tuning on sizable crowdsourcing high-quality long-turn datasets. Specifically, the Re$^3$Dial-trained model achieves better perplexity, BLEU, and ROUGE scores, while showing an improved or comparable generation diversity. In summary, these results demonstrate that Re$^3$Dial provides a well-generalized data foundation in the era of large-scale dialogue pre-training.

### 4.3.2 Human Evaluation

We conduct a pair-wise human evaluation to study the models' performance when provided with dialogue contexts of different lengths. We first randomly sample 100 long-turn contexts (consisting of at least six turns) from DuLeMon as the **Long-turn** test set. We then extract the last utterances from these contexts to form the **Short-turn** test set. We hence obtain 400 generated responses from the two models. For each pair of responses (one by the Re$^3$Dial-trained model and the other by the Original-trained model), three annotators are hired to give a preference in sensibleness and informativeness, respectively. Sensibleness mea-

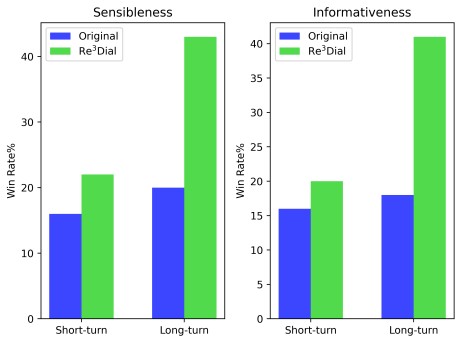

Figure 3: Pair-wise human evaluation results of the further pre-trained dialogue model. We report the win rate of each model on two test sets with different context lengths. We use Fleiss' kappa (Fleiss, 1971) to measure the inter-annotator agreement (all are moderate agreement with $0.4 \leq \kappa \leq 0.6$).

sures whether the response is relevant and consistent with the context. Informativeness measures whether the response is informative given the context. We adopt majority voting to make final decisions among three annotators. As illustrated in Figure 3, the Re$^3$Dial-trained model outperforms the baseline in both sensibleness and informativeness by a large margin on the long-turn test set. This verifies that Re$^3$Dial improves the dialogue model's ability to effectively utilize long-range context to generate more sensible and informative responses.

| Retriever | Pre-training From Scratch | | | Further Pr-training on LM | | | Further Pre-training on DM | | |
|---|---|---|---|---|---|---|---|---|---|
| | $PPL_{zero\text{-}shot}$ | BLEU-1 | BLEU-2 | $PPL_{zero\text{-}shot}$ | BLEU-1 | BLEU-2 | $PPL_{zero\text{-}shot}$ | BLEU-1 | BLEU-2 |
| Original | 193.51 | 21.81 | 10.74 | 13.06 | 17.91 | 10.41 | 59.60 | 18.82 | 9.98 |
| Random | 170.58 | 21.50 | 10.72 | 14.92 | 18.39 | 10.67 | 61.83 | 19.03 | 10.08 |
| BM25 | 192.94 | 20.61 | 10.14 | 14.69 | 18.14 | 10.58 | 73.43 | 19.48 | 10.32 |
| Contriever | 154.65 | 21.97 | 10.99 | 13.41 | 18.62 | 10.78 | 61.48 | 19.34 | 10.22 |
| Re$^3$Dial | **135.88** | **22.74** | **11.47** | **12.25** | **18.70** | **10.90** | **54.54** | **19.48** | **10.37** |

Table 3: Comparison of different retrievers. We report the average metric over three benchmarks. Cells are blue/orange if the constructed long-turn dialogue data increases/decreases the performance compared to the original baseline, respectively.

| Aspects | Irrelevance | Local Relevance | Discourse Incoherence |
|---|---|---|---|
| BM25 | 72.10 | 61.20 | 52.50 |
| Contriever | 72.20 | 66.70 | 50.90 |
| Re$^3$Dial | **97.90** | **94.90** | **68.80** |

Table 4: Accuracy of discriminating the positive retrieved session from the incoherent negative session.

| Variants | PPL↓ | Overlap↓ | Repeat Sampling↓ |
|---|---|---|---|
| Re$^3$Dial | **49.35** | **0.17** | $\mathbf{650.70_{\pm 217.33}}$ |
| w/o dialogue | 50.30 | 0.22 | $656.10_{\pm 264.64}$ |
| w/o corpus | 51.19 | 0.20 | $1,609.91_{\pm 694.91}$ |

Table 5: Effect of the diversity sampling strategy. We report the average PPL on three benchmarks. Overlap and Repeat Sampling are defined in Appendix C.

## 4.4 Analysis

**Effect of Retriever** We compare different approaches to retrieve dialogue sessions and evaluate the final dialogue model performance. We try **Random** sampling, a term-based retriever **BM25**, and a state-of-the-art dense retriever **Contriever**. Table 3 presents the results. All baselines bring fewer improvements or even inversely hurt model performance, especially zero-shot performance in the further pre-training setting. In contrast, using the retriever in Re$^3$Dial achieves consistent and significant improvements across different benchmarks and pre-training settings.

To gain a deeper understanding of the effectiveness of different retrievers in capturing global semantic and discourse relations within multi-turn dialogues, we propose to evaluate the retriever using individual tests in different aspects (Ribeiro et al., 2020). To this end, we first construct positive pairs following the strategy illustrated in Section 3.1 and introduce perturbations to create negative pairs. We then compute the retriever's accuracy in discriminating between positive and negative pairs, expecting it assigns a higher score to positive pairs. Our evaluation focuses on three aspects: Irrelevance, Local Relevance, and Discourse Incoherence. For example, to create a locally relevant negative pair, we keep one utterance from the positive session unchanged while replacing the other utterances with a randomly sampled session. More details can be found in Appendix E. The results shown in Table 4 reveal that: (1) Dense retrievers demonstrate better performance in discriminating

locally relevant negative pairs, indicating their superior ability to capture global semantic relevance. (2) Both BM25 and Contriever struggle to capture discourse coherence, showing near-random performance. (3) UDSR outperforms baselines by a large margin in capturing both global relevance and discourse coherence, verifying the effectiveness of our training data construction strategy.

Overall, these results indicate that automatically building long-turn dialogues to enhance pre-training is non-trivial. Simply improving dialogue turns is insufficient. It is important to retrieve coherent sessions based on both global semantic relevance and discourse coherence within multi-turn dialogues rather than relying solely on word overlap or semantic similarity. Otherwise, it will introduce unexpected noise or biases and lead to slightly improved or even decreased model performance.

**Effect of Diversity Sampling** To further investigate the influence of the proposed diversity sampling strategy in Re$^3$Dial, we conduct an ablation study. As shown in Table 5, the dialogue-level and corpus-level weights reduce the bias towards repetitive and generic sessions and improve the diversity and the informativeness of the constructed corpus as expected. Finally, both of them contribute to the pre-trained dialogue model's performance.

**Utilizing Long-range Context** To manifest the benefits of Re$^3$Dial, we visualize the distribution of $PPL_{zero\text{-}shot}$ on samples with varying numbers of dialogue context turns. Specifically, we first

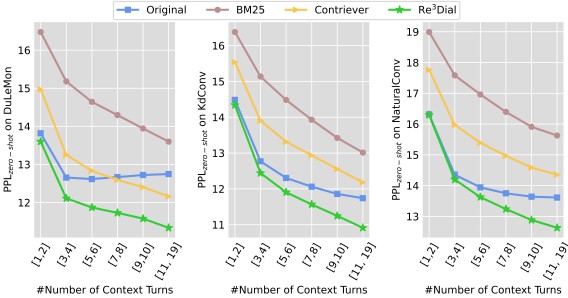

Figure 4: We report the PPL$_{\text{zero-shot}}$ varying with the number of context turns. Re$^3$Dial achieves significantly lower perplexity when given longer contexts.

select sessions from the original test set that contain at least 12 turns. We then truncate their contexts into different turns and compute the perplexity. The results shown in Figure 4 reveal that: (1) In comparison to the Original-trained model, the Re$^3$Dial-trained model achieves significantly lower perplexity as the length of context increases. Notably, when evaluating on DuLeMon, a benchmark specifically designed to evaluate the modeling of long-range dialogue history, the perplexity of the Original-trained dialogue model quickly stops decreasing after giving more than four turns of context. This indicates that pre-training on a long-turn scarce corpus restricts the model's utilization of long-range context. And Re$^3$Dial can effectively remedy this problem. (2) Although other retrieval baselines also exhibit a sharper decreasing trend in perplexity compared to the Original-trained model, they generally yield higher perplexity. This implies that while these baselines enhance the utilization of long-range context, they capture fewer long-range dependencies compared to Re$^3$Dial and may even exhibit inferior performance when the local context is more effectively utilized.

## 4.5 Comparing with Context Compression Methods

While Re$^3$Dial aims to construct a long-turn dialogue pre-training corpus to enhance the utilization of long-range context, there is another line of work that focuses on compressing long contexts into short contexts. We hence additionally conduct experiments on a retrieval-based baseline and a summarization-based baseline for long-term context modeling and compare them with Re$^3$Dial.

**Retrieval-based Context Compression** Given an original context $S = \{S_1, S_2, \cdots, S_N\}$, we use $S_N$ as the query to retrieve the top-$K$ most relevant

utterances from $\{S_1, S_2, \cdots, S_{N-1}\}$. We try two utterance retriever models: (1) Contriever: It is a state-of-the-art dense retriever model. (2) Sentence-BERT (Reimers and Gurevych, 2019): It is an encoder model fine-tuned for sentence similarity. We set $K = 2$ in our experiments.

**Summarization-based Context Compression** We introduce an additional summarization model to summarize long-term context into short sentences. We try two summarization models: (1) Pegasus-523M (Zhang et al., 2020): It is a widely-adopted encoder-decoder model specifically pre-trained and fine-tuned for text summarization. (2) ChatGLM-66B (Zeng et al., 2022): It is a widely-adopted instruction-tuned large language model.

We report the average PPL$_{\text{zero-shot}}$ over three multi-turn dialogue benchmarks. From the results shown in Table 6, we observe that Re$^3$Dial significantly outperforms all baselines in long-turn dialogue benchmarks. Moreover, augmenting the dialogue model with a context summarization model or a retriever shows less improvement or inversely hurts model performance in several cases.

On the one hand, the two-stage framework suffers from error propagation due to the introduced summarization model or the retriever. For example, both the summarization model and the retriever may lose important information in the original context. Moreover, the summarization model could also suffer from hallucination problems (Maynez et al., 2020), thereby introducing new noises. In contrast, Re$^3$Dial keeps the original long-turn context unchanged and thus does not lead to information loss or introduce new noises. On the other hand, we conjecture that augmenting dialogue models with the context summarization model requires further training on summarization-based dialogue datasets (Xu et al., 2022a). In contrast, Re$^3$Dial does not require collecting additional training datasets and greatly improves the model performance.

## 4.6 Case Study

As shown in Table 7, the Original-trained model mainly focuses on local context and tends to generate more generic responses (e.g., "I think the same" in responding to the preceding utterance, "they thought it was too risky"). In contrast, the Re$^3$Dial-trained dialogue model generates words related to the long-range context (e.g., "*fashion designer*" which has been mentioned nine turns prior),

| Method | Pre-training From Scratch | Further Pre-training on LM | Further Pre-training on DM |
|---|---|---|---|
| Original | 193.51 | 13.06 | 59.60 |
| + Retrieval (Sentence-BERT) | 169.36 | 13.87 | 65.17 |
| + Retrieval (Contriever) | 168.85 | 13.69 | 64.03 |
| + Summarization (Pegasus-523M) | 172.21 | 14.20 | 68.16 |
| + Summarization (ChatGLM-66B) | 182.52 | 13.79 | 63.48 |
| Re$^3$Dial | **135.88** | **12.25** | **54.54** |

Table 6: Comparison between Re$^3$Dial and retrieval-based and summarization-based context compression methods. We report the average PPL over three benchmarks. Cells are blue/orange if the method increases/decreases the performance compared to the original baseline, respectively.

---

**Dialogue Context:**
**A:** You're about to graduate from the Central Academy of Fine Arts, aren't you?
**B:** Yes, I'll be a 4th year undergraduate.
**A:** What do you want to do after graduating?
**B:** My dream is to be a fashion designer , so that's definitely what I'll be doing after graduating.
**A:** Being a designer is difficult because it requires inspiration.
**B:** I know it is difficult. I could get inspiration from Dunhuang murals or ethnic costumes.
**A:** So do you want to design Chinese-style clothes?
**B:** Yes, I think this kind of clothes look more mysterious and have a sense of history.
**A:** So are you planning to open a studio?
**B:** I've always had this idea, but I don't have enough money.
**A:** Why don't you find a partner?
**B:** I tried, but my classmates they didn't want to do it, they thought it was too risky.

**Original**: Well, actually I think the same.

**Re$^3$Dial**: I think it would be a good idea to find another experienced fashion designer , which will help you to achieve your dream.

Table 7: Generated responses from the model pre-trained on Re$^3$Dial and Original corpus (translated from Chinese to English). We highlight the generated spans that are related to long-range context.

leading to a more sensible and specific response.

### 4.7 Constructing Re$^3$Dial-1B

To show the efficiency of constructing large-scale long-turn dialogue data with Re$^3$Dial and allow researchers to explore Re$^3$Dial easily, we finally release Re$^3$Dial-1B, an improved corpus based on the original EVA corpus that contains 1B sessions with 11.3 turns on average ($5\times$ longer than that of the original EVA corpus). The whole pipeline costs about five days with 32 V100 32G GPUs.

### 5 Conclusion

This paper presents Re$^3$Dial, a framework that automatically builds billion-scale long-turn dialogues by reorganizing existing short-turn ones, thereby enhancing the model's ability to utilize long-range context from a data-centric perspective. Re$^3$Dial leverages a dense retriever trained on massive unlabeled dialogues to improve the coherence of concatenated sessions. Furthermore, a diversity sampling strategy is proposed to penalize repetitive or generic sessions, improving the informativeness and diversity of the constructed corpus. Extensive experiments demonstrate that Re$^3$Dial significantly improves the model's performance on various multi-turn dialogue benchmarks across different pre-training settings due to the better utilization of long-range contexts. Finally, we provide a toolkit for efficiently rescaling conversations with Re$^3$Dial and successfully build Re$^3$Dial-1B, a large-scale long-turn dialogue corpus that contains 1B Chinese dialogues with 11.3 turns on average. Our work provides a new data foundation for building large-scale pre-trained dialogue models.

### Limitations

Although we have already verified the effectiveness of Re$^3$Dial using UDSR, there are still several directions to further improve the retrieval performance. For instance, while we explore using the BM25 hard negatives in our experiments, there are more advanced negative sampling strategies (Xiong et al., 2020). We will explore these directions and further improve UDSR.

Besides, despite that Re$^3$Dial can be adapted to any open-domain dialogue corpus in any language, we currently only conduct experiments based on a Chinese open-domain dialogue corpus. It is necessary to further collect other language dialogue corpus, such as the English dialogue data from Reddit, and verify the effectiveness of Re$^3$Dial.

Moreover, we note that our work just makes the first step to automatically construct a long-turn dialogue corpus for enhancing long-turn dialogue pre-training. Based on the Re$^3$Dial framework, future

works could further explore: (1) can we flexibly control the conversation flow to fit the specific characteristics in real long-turn dialogues (e.g., topic-drift) by adjusting the degree of coherence? (2) can we design pre-training tasks to utilize the additional signals in the constructed long-turn dialogue corpus, e.g., the similarity score?

## Ethics Statement

Our experiments are conducted based on the existing web-crawled dialogue corpus. Despite that the corpus has been preprocessed for safety concerns (e.g., filtering sensitive words) (Zhou et al., 2021), Xu et al. (2020) show that the pre-trained dialogue models still have unsafe behaviors, such as generating toxic responses. Therefore, dialogue models should be carefully examined before being made publicly available.

We hire three annotators from a professional data annotation company. We do not ask about any private information in the annotation process. We pay each annotator 0.2$ for comparing each pair of retrieving results. Each comparison costs about 1 minute on average, so the payment is quite reasonable.

## Acknowledgements

This work was supported by the National Key Research and Development Program of China (No. 2021ZD0113304), the National Science Foundation for Distinguished Young Scholars (with No. 62125604) and the NSFC projects (Key project with No. 61936010).

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

# A Data Information

## A.1 Retriever Data

We first derive a subset of the original EVA pre-training corpus that consists of dialogues with more than four turns. We then randomly sample 1,000,000/49,000/1,000 from this subset for the train/validation/test set, respectively. We use the in-batch negative trick in our experiment (Karpukhin et al., 2020). Table 8 shows the statistics of the retriever data.

| Split | # Num Examples | In-Batch | # Num Negatives |
|---|---|---|---|
| **Train** | 1,000,000 | ✔ | 1023 + 1024 |
| **Valid** | 49,000 | ✔ | 1023 + 1024 |
| **Test** | 1,000 | ✗ | 999 + 0 |

Table 8: Statistics of retriever data. **# Num Examples** denotes the number of query sessions. **In-Batch** denotes whether use the trick of in-batch negatives, which we use in the training stage of UDSR. **# Num Negatives** denotes the number of negatives for each query session, which consists of two parts, i.e., the number of random negatives and the number of BM25 negatives.

## A.2 Dialogue Pre-training Data

We randomly sample 5 million dialogue sessions for experiments and 1 billion dialogue sessions for constructing Re$^3$Dial-1B from the original EVA pre-training corpus.

## A.3 Dialogue Benchmarks

Table 9 shows the data statistics of the three benchmarks. We use the official split of DuLeMon and

KdConv. For NaturalConv, considering that the original training set is too large (containing nearly 20,000 sessions), we randomly sample 5,000 from it as the training set. Moreover, as we focus on multi-turn dialogue modeling instead of grounding dialogue in this paper, we only use plain dialogue data for training and evaluation, leaving out any grounding information, such as knowledge, persona, or document.

## B Training Details

In this section, we provide more training details of our experiment.

For retriever training, we concatenate all utterances within a multi-turn dialogue with the <SEP> token. We set the maximum sequence length to 256, batch size to 512, the initial learning rate of AdamW optimizer to 5e-5. The best checkpoint is selected based on the Top-1 recall on the validation set.

For dialogue pre-training, we concatenate all utterances within a multi-turn dialogue with the <SEP> token. We use the Noam scheduler to adjust the learning rate. We set the maximum sequence length to 256 for decoder-only models and 256 + 64 for encoder-decoder models. For the setting of pre-training from scratch, we set the batch size to 512, the initial learning rate of AdamW optimizer to 1e-2, and the warmup step to 1K. Due to the limited computational resources and time, we pre-train the 6B non-causal decoder model for 10K steps. For the setting of further pre-training on LM, we set the batch size to 256, the initial learning rate of AdamW optimizer to 1e-4, and the warmup step to 100. We pre-train GPT2-small[4] for 30K steps. For the setting of further pre-training on DM, we set the batch size to 128, the initial learning rate of AdamW optimizer to 1e-5, the gradient accumulation steps to 2, and the warmup step to 100. We pre-train ChatGLM (Zeng et al., 2022)[5] for 30K steps.

For dialogue fine-tuning, we keep the same maximum sequence length with pre-training. We manually select the batch size from [32, 64, 128] and the initial learning rate of AdamW optimizer from [1e-4, 5e-5] based on the perplexity on the validation set. We generate responses using beam search (Holtzman et al., 2019).

---

[3]https://huggingface.co/uer/gpt2-chinese-cluecorpussmall

[4]https://huggingface.co/THUDM/chatglm-6b

To improve the training efficiency, we adopt the mixed-precision training (Micikevicius et al., 2017), gradient checkpointing, and ZeRO (Rajbhandari et al., 2020) implemented in DeepSpeed (Rasley et al., 2020).

It costs about 72 hours to train UDSR on 1 million examples. It costs about 15~48 hours for dialogue pre-training and 0.5~4 hours for dialogue fine-tuning, depending on different backbone models and downstream benchmarks.

## C More Metrics

**Overlap Score** For each utterance in a multi-turn dialogue, it is computed as the length of the LCS between the utterance and its context divided by the length of the utterance. And we use micro-averaging to derive the overlap score of the overall constructed corpus.

**Repeated Sampleing** Let $c_i$ be the sampled times of the dialogue $S_i$ after finishing contructing the corpus. It measures the mean and the standard deviation of the sampled times of the Top-$K$ sessions among all the corpus sorted by $c_i$. We use $K = 1000$ in our experiments.

## D Experiments on English Benchmarks

Our proposed Re[3]Dial is a language-agnostic framework. To further verify the effectiveness of Re[3]Dial in different languages, we conduct experiments on an English corpus. Specifically, we collect an English dialogue pre-training corpus, which consists of 1 million conversations from Reddit. The average number of turns in the original corpus is 2.3.

We then conduct further pre-training on a GPT-2 large model. We report PPL$_{\text{zero-shot}}$ on three widely-adopted English multi-turn dialogue benchmarks, including Blended Skill Talk (Smith et al., 2020), PersonaChat (Zhang et al., 2018), and Wizard of Wikipedia (Dinan et al., 2019). From the results shown in Table 10, we can see that Re[3]Dial achieves significantly lower PPL than the Original baseline. These results demonstrate the effectiveness of Re[3]Dial in English.

## E Analysis of the Retriever

### E.1 Constructing Incoherent Examples

Given a human-written $K$-turn dialogue session $S = \{u^1, u^2, \cdots, u^K\}$, we first construct the positive pair $(S_q, S_c^+)$ as follows:

| Benchmark | Train | | Valid | | Test | |
|---|---|---|---|---|---|---|
| | # Session | Avg. # Turn | # Session | Avg. # Turn | # Session | Avg. # Turn |
| DuLeMon | 2,401 | 16.2 | 300 | 16.0 | 300 | 16.1 |
| KdConv | 3,600 | 18.5 | 450 | 20.7 | 450 | 21.6 |
| NaturalConv | 5,000 | 20.1 | 272 | 20.1 | 272 | 20.1 |

Table 9: Statistics of three multi-turn open-domain dialogue benchmarks.

| Pre-training Data | Blended Skill Talk | PersonaChat | Wizard of Wikipedia |
|---|---|---|---|
| **Original** | 33.72 | 39.61 | 50.05 |
| **Re$^3$Dial** | **29.17** | **31.37** | **48.62** |

Table 10: PPL$_{\text{zero-shot}}$ in the setting of further pre-training on LM on three English multi-turn dialogue benchmarks. The best performance is highlighted in **bold**.

$$S_q = \{u^4, u^5, \cdots, u^{K-3}\}$$
$$S_c^+ = \{u^{K-2}, u^{K-1}, u^K\}$$

We then introduce the following perturbations to create the negative consecutive session $S_c^-$ in the specific aspect:

- **Irrelevance:** we create $S_c^-$ by randomly sampling a session.

- **Local Irrelevance**: we create $S_c^-$ by maintaining one single utterance in $S_c^+$ unchanged while replacing the other utterances with a randomly sampled session.

- **Discourse Incoherence** we create $S_c^-$ by using the first three utterances in $S$, i.e., $S_c^- = \{u^1, u^2, u^3\}$.

### E.2 Automatic Evaluation

Table 11 shows the automatic evaluation results of the retriever. We can see that UDSR outperforms baselines by a large margin. Moreover, we also test two ablated versions of UDSR in encoding strategies, which either encode the last utterance of $S_q$ or encode the first utterance of $S_c$ for retrieving. The results further demonstrate the importance of capturing global features in this task. We also study the inference speed of the retriever since we aim to build a billion-scale pre-training corpus. We can see that with the help of GPU and FAISS, dense retrievers can achieve incredible efficiency, processing 192 query sessions per second with a single V100 GPU. In contrast, BM25 (implemented using ElasticSearch) can only process 22 query sessions per second.

| Retriever | Top-5 | Top-20 | Speed |
|---|---|---|---|
| **BM25** | 38.04 | 47.75 | 22 it/s |
| **Contriever** | 49.60 | 62.40 | 192 it/s |
| **UDSR** | **79.70** | **89.70** | **192 it/s** |
| w/ $S_q$(**single**) | 59.10 | 70.80 | 192 it/s |
| w/ $S_c$(**single**) | 59.90 | 74.60 | 192 it/s |

Table 11: Automatic evaluation results of different retrievers. We report the Top-$k$ recall and the inference speed for retrieving the Top-10 sessions from 5M sessions.

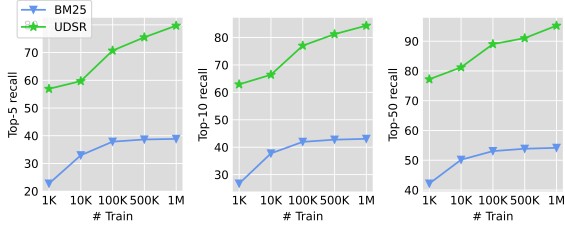

Figure 5: Top-$k$ recall varying with different numbers of training examples.

### E.3 Sample Efficiency

We investigate how the retriever's performance varies with the number of training examples. As shown in Figure 5, UDSR significantly outperforms BM25, starting from just 1K training examples. Moreover, UDSR is more data-hungry as a dense retrieval method (Xiong et al., 2020; Karpukhin et al., 2020). While the performance of BM25 is close to convergence with 100K training examples, the performance of UDSR is still markedly improved. Considering that there is massive unlabeled plain dialogue data, the retrieval performance of UDSR might be further improved with more training examples.

| # Context Turns | Top-5 | Top-20 |
|---|---|---|
| 3 | 61.60 | 77.00 |
| 6 | 69.20 | 83.40 |
| 9 | 72.20 | 86.30 |

Table 12: We report the Top-$k$ recall varying with the number of context turns.

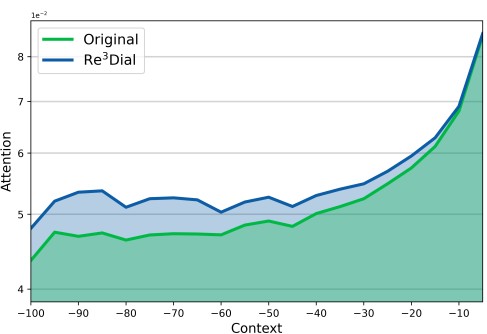

Figure 6: Attention weights on the context (in log-scale) in the final layer of the dialogue model pre-trained on the Original corpus and Re³Dial averaged on the test set of DuLeMon.

### E.4 Effect of Context Length

While we only use the last session as the query for retrieval to reduce the time cost of large-scale retrieval, we are interested in how the retrieval performance of our UDSR varies with different context lengths. We sample 1000 long-turn (at least 12-turn) dialogues from Weibo data, where the last 3-turn serves as the $S_c$. We then investigate how the UDSR's performance varies with the number of turns of $S_q$. As shown in Table 12, UDSR can effectively use longer context and achieve better retrieval performance.

### F Attention Weights Distribution

We visualize the attention weights distribution on the context tokens when predicting the next target token in Figure 6. For each token, we select the Top-5 context tokens with the largest attention scores. We then take the average attention score in each interval of five tokens. Compared with the original short-turn pre-training corpus, pre-training on Re³Dial achieves better awareness of the older context. The result indicates that pre-training on the automatically constructed long-turn dialogue data with Re³Dial learns to better attend to and utilize long-range context for response generation.

### G Examples of Re³Dial

We present two examples of automatically constructed long-turn dialogues by Re³Dial in Table 13

### H Instructions for Human Evaluation

We show our instructions for human evaluation in Figure 7.

**Automatically Constructed Long-turn Dialogue**

*Example #1*

**A**: I really hate to say "okay, its my fault" during an argument.· · · This attitude of unwillingness to solve the problem only make things words · · ·

**B**: Communication is the best way to maintain a good relationship.

**A**: · · · Even you have different views, don't immediately dismiss other's views, but first agree with part of it· · · Please take care to have a good attitude when communicating· · ·

**B**: That's true. I want to communicate properly, and I don't want to fight.· · ·

**A**: The best way for two people to get along is through both arguments and warmth.· · · Sometimes your temper is like a sword· · · it can hurt someone who cares about you· · ·

**B**: A good relationship leads to a quick make-up after a fight.· · ·

**A**: Actually, many times I would first apologize. This is not because I was wrong, but because I cherish you· · · I know that I have to take care of your feelings· · ·

**B**: That's why we say that the one who loves the most is the one who is most humble.

**A**: I think a good relationship is to love each other equally deeply.

**B**: It's important that girls should not be to humble in a relationship, or you will really despise yourself in the end. Loving someone is mutual.

⋮

*Example #2*

**A**: Parents really expect too much from their children in this society, and such children are 100% mentally unhealthy even if they achieve success in the future.

**B**: There is no need for children to start working so hard from such a young age.

**A**: Parents want to send their children to an Ivy League School, but they just spend time on playing phones. It is only a pipe dream to expect children to work hard without parents themselves set an example.

**B**: Honestly speaking, it is hard to practice what you preach for educating children.

**A**: Every time this topic of 'tutoring your child's homework' comes up, it has a lot of resonance. I think that the root of the problem is not how well the child learns, but what is the mindset of the parents. When parents want to get angry, they should ask themselves two questions: 1. Can you do what you are asking your child to do when you were this age? 2. Even if you could have done it, why should your child have to do it?

**B**: I agree. You can't blame others but yourself for your child's IQ.

**A**: · · · Most of what we experienced as children was percussive education· · · We should encourage our children more· · ·

**B**: Yes, recognizing your child's efforts should precede criticizing his shortcomings.

⋮

Table 13: Examples of the automatically constructed long-turn dialogue by Re³Dial, which are translated from Chinese to English.

| Labeling instructions | | | |
|---|---|---|---|
| This study aims to evaluate dialogue generation systems. Specifically, for each dialogue context, the dialogue generation system will generate a response.

In the process of evaluation, you will be given on dialogue context and two generated sessions by two difference dialogue generation systems. You need to compare them in terms of sensibleness and informativeness, and give a preference (Win, Lose, Tie).

By **sensibleness**, we mean that the generated response should be semantically relevant and consistent with the context.
By **informativeness**, we mean that the generated response should be informative given the context. | | | |
| | Examples | Sensibleness | Informativeness |
| Context | A: You're about to graduate from the Central Academy
of Fine Arts, aren't you?
B: Yes, I'll be a 4th year undergraduate.
A: What do you want to do after graduating?
B: My dream is to be a fashion designer , so that's definitely what I'll be doing after graduating.
A: Being a designer is difficult because it requires inspiration.
B: I know it is difficult. I could get inspiration from Dunhuang murals or ethnic costumes.
A: So do you want to design Chinese-style clothes?
B: Yes, I think this kind of clothes look more mysterious and have a sense of history.
A: So are you planning to open a studio?
B: I've always had this idea, but I don't have enough money.
A: Why don't you find a partner?
B: I tried, but my classmates they didn't want to do it, they thought it was too risky | Response 2
Win | Response 2
Win |
| Generated Response 1 | B: Well, actually I think the same. | | |
| Generated Response 2 | B: I think it would be a good idea to find another experienced fashion designer, which will help you to achieve your dream. | | |

Figure 7: Instructions given to labelers for human evaluations.