# OpenReview forum: "Re$^3$Dial: Retrieve, Reorganize and Rescale Conversations for Long-Turn Open-Domain Dialogue Pre-training"
_EMNLP/2023/Conference — EMNLP 2023 Main_

### Official Review · Reviewer_oPUQ · 2023-08-06

**Soundness:** 3

**Excitement:**

4: Strong: This paper deepens the understanding of some phenomenon or lowers the barriers to an existing research direction.

**Paper Topic And Main Contributions:**

The objective of this paper is to introduce the Re3Dial framework, which addresses the issue of pre-trained dialogue models lacking the ability to effectively utilize long-range context. To overcome this limitation, the framework automatically constructs billion-scale long-turn dialogues by implementing conversation retrieval, reorganization, and rescaling techniques. The Re3Dial approach consists of four steps for the automatic construction of long-turn dialogues. Firstly, the constructed session is initialized using the original session. Then, a retrieval system called Unsupervised Dense Session Retriever is employed to retrieve the top-K coherent sessions from the dialogue corpus. Subsequently, diversity sampling is applied to select a consecutive session. Finally, the constructed session is updated and extended by concatenating the sampled session. This process is repeated for a specified number of turns, controlling the length of the constructed dialogue. Experimental results demonstrate that the proposed framework significantly enhances the dialogue model's ability to generate coherent and informative responses in multi-turn dialogues.

**Questions For The Authors:**

This paper introduces a general framework and builds its experimental setup incorporating multiple models. However, it fails to compare any additional baseline methods, such as alternative retrieval enhancements or different approaches to long-term modeling.

**Reasons To Accept:**

This paper proposes a Re3Dial conversation framework that automatically constructs billion-scale long-turn dialogues by implementing conversation retrieval, reorganization, and rescaling techniques. The motivation behind this approach is to enhance the dialogue model's ability to utilize long-range context and generate more sensible and informative responses in multi-turn dialogue. It sounds reasonable.

**Reasons To Reject:**

This paper introduces a general framework and builds its experimental setup incorporates multiple models. However, it fails to compare any additional baseline methods, such as alternative retrieval enhancements or different approaches to long-term modeling. From my perspective, as an applied article, this is not reasonable. Furthermore, the overall method lacks substantial innovation.

**Reproducibility:**

3: Could reproduce the results with some difficulty. The settings of parameters are underspecified or subjectively determined; the training/evaluation data are not widely available.

**Reviewer Confidence:**

3: Pretty sure, but there's a chance I missed something. Although I have a good feel for this area in general, I did not carefully check the paper's details, e.g., the math, experimental design, or novelty.

---

> ### Author Rebuttal · Authors · 2023-08-28
>
> > 1. It fails to compare any additional baseline methods, such as alternative retrieval enhancements or different approaches to long-term modeling.
>
> We additionally conduct experiments on a retrieval-based baseline and a summarization-based baseline for long-term context modeling.
>
> For the retrieval-based baseline, given an original context $S = {S_1, S_2, ..., S_N}$, we use $S_N$ as the query to retrieve the top-$K$ most relevant utterances from $\{S_1, ..., S_{N-1}\}$. We try two utterance retriever models: (1) Contriever: It is a state-of-the-art dense retriever model. (2) Sentence-BERT [1]: It is an encoder model fine-tuned for sentence similarity. We set $K=2$ in our experiments.
>
> For the summarization-based baseline, we introduce an additional summarization model to summarize long-term context into short sentences. We try two summarization models: (1) Pegasus-523M [2]: It is a widely-adopted encoder-decoder model specifically pre-trained and fine-tuned for text summarization. (2) ChatGLM-66B [3]: It is a widely-adopted instruction-tuned large language model.
>
> We report the average PPL$_{zero-shot}$ over three multi-turn dialogue benchmarks. From the results shown in the following table, we observe that Re$^3$Dial significantly outperforms all baselines in long-turn dialogue benchmarks. Moreover, augmenting the dialogue model with a context summarization model or a retriever shows less improvement or inversely hurts model performance in several cases.
>
> On the one hand, this two-stage framework suffers from error propagation due to the introduced summarization model or the retriever. For example, both the summarization model and the retriever may lose important information in the original context. Moreover, the summarization model could also suffer from hallucination problems [4], thereby introducing new noises. In contrast, Re$^3$Dial keeps the original long-turn context unchanged and thus does not lead to information loss or introduce new noises.
>
> On the other hand, we conjecture that augmenting dialogue models with the context summarization model requires further training on summarization-based dialogue datasets [5]. In contrast, Re$^3$Dial does not require collecting additional training datasets and greatly improves the model performance.
>
> | Method                         | Pre-training From Scratch | Further Pre-training on LM | Further Pre-training on DM |
> | ------------------------------ | ------------------------- | -------------------------- | -------------------------- |
> | Original                       | 193.51                    | 13.06                      | 59.60                      |
> | + Retrieval (Sentence-BERT)    | 169.36                    | 13.87                      | 65.17                      |
> | + Retrieval (Contriever)       | 168.85                    | 13.69                      | 64.03                      |
> | + Summarization (Pegasus-523M) | 172.21                    | 14.20                      | 68.16                      |
> | + Summarization (ChatGLM-66B)  | 182.52                    | 13.79                      | 63.48                      |
> | Re$^3$Dial                     | **135.88**                | **12.25**                  | **54.54**                  |
>
>
> > 2. The overall method lacks substantial innovation.
>
> At its core, our proposed Re$^3$Dial framework provides a novel perspective to alleviate the scarcity of long-turn conversations, i.e., automatically building a billion-scale long-turn dialogue corpus via concatenating existing short-turn dialogue.
>
> However, naive implementations of this novel idea may result in incoherent, low-diversity, or low-informativeness long-turn dialogue corpus. Therefore, our novelty also lies in addressing the following two challenges:
>
> - **Preserving Dialogue Coherence**: We build a high-performance dialogue session retriever by training on massive unlabeled dialogues, which are readily available in various languages. Additionally, we propose to use individual tests to evaluate the effectiveness of the dialogue session retriever in capturing global semantic and discourse relations within multi-turn dialogues.
>
> - **Enhancing Dialogue Diversity**: We propose two "inherent problems" of the dialogue pre-training corpus, which may decrease diversity during concatenation. And our proposed diversity sampling strategy effectively tackles these concerns.
>
> Extensive experiments demonstrate that Re$^3$Dial provides a well-generalized data foundation in the era of large-scale dialogue pre-training.
>
> [1] Sentence-BERT: Sentence Embeddings using Siamese BERT-Networks. EMNLP2019
>
> [2] Pegasus: pre-training with extracted gap-sentences for abstractive summarization. ICML2020
>
> [3] GLM-130B: An Open Bilingual Pre-trained Model. ICLR2023
>
> [4] On Faithfulness and Factuality in Abstractive Summarization. ACL2020
>
> [5] Beyond goldfish memory long-term open-domain conversation. ACL2022

---

### Official Review · Reviewer_RkmQ · 2023-08-10

**Soundness:** 4

**Excitement:**

4: Strong: This paper deepens the understanding of some phenomenon or lowers the barriers to an existing research direction.

**Paper Topic And Main Contributions:**

The paper proposes Re3Dial, a technique to augment dialogue data by creating longer dialogues from the shorter ones. For that, they use a dense retriever to select coherent dialogue segments for combining, and a diversity sampling procedure to increase variability in the dialogue trajectories.
The model trained (depending on the setup, it's either a from-scratch model or GPT2-small or ChatGLM) on the resulting augmented 1-billion dialogue dataset outperforms the original model trained on EVA corpus in both automatic metrics on 3 Chinese dialogue datasets - and human metrics on one of them.

The authors also release the resulting dialogue dataset with 1B dialogues in Chinese with the average length of 11.3 turns.

**Reasons To Accept:**

* a novel and effective technique for dialogue data augmentation is proposed - it addresses the problem of composing longer dialogues of shorter segments while preserving coherence and maximizing diversity of conversations
* training on such a dataset results in improved dialogue system's performance both on automatic and human metrics
* the authors contribute the resulting 1B dialogue dataset for public use

**Reasons To Reject:**

N/A

**Reproducibility:**

4: Could mostly reproduce the results, but there may be some variation because of sample variance or minor variations in their interpretation of the protocol or method.

**Reviewer Confidence:**

4: Quite sure. I tried to check the important points carefully. It's unlikely, though conceivable, that I missed something that should affect my ratings.

---

> ### Author Rebuttal · Authors · 2023-08-28
>
> Thanks for your positive review!

---

### Official Review · Reviewer_YxhS · 2023-08-11

**Soundness:** 4

**Excitement:**

4: Strong: This paper deepens the understanding of some phenomenon or lowers the barriers to an existing research direction.

**Paper Topic And Main Contributions:**

This paper addresses a highly significant issue in the field of open-domain chatbots, which involves extending the current short-turn open-domain dialogue corpus into a long-turn corpus. Moreover, the methodology proposed in this paper, including Re^3Dial, is very effective due to its automated approach. Also, they have a plan to release 1B corpora as open-source data.

**Reasons To Accept:**

- This paper highlights critical issues in the current open-domain chatbot domain concerning the utilization of corpora and proposes methodologies to address them.

- Furthermore, it defines inherent problems that might arise during the resolution process and provides clear solutions to these issues. The overall process is also well-structured.

- The proposed methodology appears to be generally applicable to corpora based on different languages as well.

- The paper effectively demonstrates the efficacy of the augmented corpus through experiments, and the results are also impressive.

**Reasons To Reject:**

- While the methodology proposed in this paper appears to be language agnostic, the experiments are conducted only on Chinese corpora, so it cannot guarantee proper functioning across different languages.

- The methodologies used within the framework presented in the paper are not significantly novel.

**Reproducibility:**

4: Could mostly reproduce the results, but there may be some variation because of sample variance or minor variations in their interpretation of the protocol or method.

**Reviewer Confidence:**

5: Positive that my evaluation is correct. I read the paper very carefully and I am very familiar with related work.

---

> ### Author Rebuttal · Authors · 2023-08-28
>
> > 1. While the methodology proposed in this paper appears to be language agnostic, the experiments are conducted only on Chinese corpora, so it cannot guarantee proper functioning across different languages.
>
> We conduct further pre-training on a GPT-2 large model. We report PPL$_{zero-shot}$ on three widely-adopted English multi-turn dialogue benchmarks, including Blended Skill Talk [1], PersonaChat [2], and Wizard of Wikipedia [3]. From the results shown in the following table, we can see that Re$^3$Dial achieves significantly lower PPL than the Original baseline. These results demonstrate the effectiveness of Re^3Dial in English.
> | Pre-training Data | Blended Skill Talk | PersonaChat | Wizard of Wikipedia |
> | ----------------- | ------------------ | ----------- | ------------------- |
> | Original          | 33.72              | 39.61       | 50.05               |
> | Re$^3$Dial        | 29.17              | 31.37       | 48.62               |
>
> > 2. The methodologies used within the framework presented in the paper are not significantly novel.
>
> At its core, our proposed Re$^3$Dial framework provides a novel perspective to alleviate the scarcity of long-turn conversations, i.e., automatically building a billion-scale long-turn dialogue corpus via concatenating existing short-turn dialogue.
>
> However, naive implementations of this novel idea may result in incoherent, low-diversity, or low-informativeness long-turn dialogue corpus. Therefore, our novelty also lies in addressing the following two challenges:
> - **Preserving Dialogue Coherence**: We build a high-performance dialogue session retriever by training on massive unlabeled dialogues, which are readily available in various languages. Additionally, we propose to use individual tests to evaluate the effectiveness of the dialogue session retriever in capturing global semantic and discourse relations within multi-turn dialogues.
> - **Enhancing Dialogue Diversity**: We propose two "inherent problems" of the dialogue pre-training corpus, which may decrease diversity during concatenation. And our proposed diversity sampling strategy effectively tackles these concerns.
>
> Extensive experiments demonstrate that Re$^3$Dial provides a well-generalized data foundation in the era of large-scale dialogue pre-training.
>
>
> [1] Can You Put it All Together: Evaluating Conversational Agents’ Ability to Blend Skills. ACL2020
>
> [2] Personalizing Dialogue Agents: I have a dog, do you have pets too? ACL2018
>
> [3] Wizard of Wikipedia: Knowledge-Powered Conversational agents. ICLR2019

---

### Official Review · Reviewer_2y1L · 2023-08-11

**Typos Grammar Style And Presentation Improvements:** n/a
**Soundness:** 4

**Excitement:**

3: Ambivalent: It has merits (e.g., it reports state-of-the-art results, the idea is nice), but there are key weaknesses (e.g., it describes incremental work), and it can significantly benefit from another round of revision. However, I won't object to accepting it if my co-reviewers champion it.

**Missing References:**

n/a

**Paper Topic And Main Contributions:**

This paper propose a data augmentation framework to generate long conversations by retrieving multiple relevant short conversations and reorganize them into one long conversation.

**Questions For The Authors:**

For the pre-training experiment, how many examples do you use from the Re^3Dial dataset? It's mentioned 5 million examples from EVA corpus are used, is the number the same or larger for Re^3Dial dataset?

**Reasons To Accept:**

- The methodology and experiment are well explained.
- The idea of retrieve and reorganize to generate long conversations might be useful under some circumstances.

**Reasons To Reject:**

- The marginal improvements might not be worth the extra efforts. The Re^3Dial dataset has 5x utterances, but the improvement is marginal. Also, it's unclear if they actually use more conversation from the Re^3Dial dataset.

**Reproducibility:**

4: Could mostly reproduce the results, but there may be some variation because of sample variance or minor variations in their interpretation of the protocol or method.

**Reviewer Confidence:**

3: Pretty sure, but there's a chance I missed something. Although I have a good feel for this area in general, I did not carefully check the paper's details, e.g., the math, experimental design, or novelty.

---

> ### Author Rebuttal · Authors · 2023-08-28
>
> > 1. The Re$^3$Dial dataset has 5x utterances. Also, it's unclear if they actually use more conversation from the Re$^3$Dial dataset.
>
> We use 5 million examples for Re$^3$Dial during pre-training in our experiments, which is the same number as other baselines instead of 5X utterances. Specifically, for each response in the original corpus, Re$^3$Dial constructs a longer context for it. And the loss is only computed on the response tokens.
>
> > 2. The marginal improvements might not be worth the extra efforts.
>
> We would like to further clarify the substantial improvements of Re$^3$Dial.
>
> Firstly, for automatic evaluations (Section 4.3.1), Re$^3$Dial demonstrates substantial improvement in automatic metrics in an open-ended generation task (i.e., open-domain dialogue generation in our experiments). Let us reference the results in DuLeMon [1]. It was observed that increasing model parameters from 110M to 1.6B leads to an improvement of 0.0/0.6 in terms of BLEU-1/2, respectively. In this light, the performance boost achieved by Re$^3$Dial is substantial. Furthermore, it is essential to underline that Re$^3$Dial outperforms the Original baseline in terms of PPL$_{zero-shot}$ and PPL by a large margin. As pointed out by [2][3][4], perplexity shows a stronger correlation with human judgments than other automatic metrics (e.g., BLEU).
>
> Secondly, considering that automatic metrics cannot fully demonstrate the performance of open-ended dialogue models [2][3][4][5], we further conduct human evaluations (Section 4.3.2). As illustrated in Figure 3, the Re$^3$Dial-trained dialogue model performs significantly better than the Original baseline in sensibleness and coherence in the long-turn test set.
>
>
> [1] Long Time No See! Open-Domain Conversation with Long-Term Persona Memory. ACL2022 findings.
>
> [2] Towards a Human-like Open-Domain Chatbot. arXiv2020
>
> [3] How not to evaluate your dialogue system: An empirical study of unsupervised evaluation metrics for dialogue response generation. EMNLP2016
>
> [4] Towards an Automatic Turing Test: Learning to Evaluate Dialogue Responses. ACL2017
>
> [5] RUBER: An Unsupervised Method for Automatic Evaluation of Open-Domain Dialog Systems. AAAI2018

---

### Official Review · Reviewer_dkfb · 2023-08-12

**Soundness:** 4

**Excitement:**

3: Ambivalent: It has merits (e.g., it reports state-of-the-art results, the idea is nice), but there are key weaknesses (e.g., it describes incremental work), and it can significantly benefit from another round of revision. However, I won't object to accepting it if my co-reviewers champion it.

**Missing References:**

(1) While all the models under the pre-training setting 'Further Pre-training on DM' are built upon ChatGLM-6B, the authors did not cite the following two papers relevant to ChatGLM-6B. Please consider citing them properly.

@article{zeng2022glm,
  title={Glm-130b: An open bilingual pre-trained model},
  author={Zeng, Aohan and Liu, Xiao and Du, Zhengxiao and Wang, Zihan and Lai, Hanyu and Ding, Ming and Yang, Zhuoyi and Xu, Yifan and Zheng, Wendi and Xia, Xiao and others},
  journal={arXiv preprint arXiv:2210.02414},
  year={2022}
}

@inproceedings{du2022glm,
  title={GLM: General Language Model Pretraining with Autoregressive Blank Infilling},
  author={Du, Zhengxiao and Qian, Yujie and Liu, Xiao and Ding, Ming and Qiu, Jiezhong and Yang, Zhilin and Tang, Jie},
  booktitle={Proceedings of the 60th Annual Meeting of the Association for Computational Linguistics (Volume 1: Long Papers)},
  pages={320--335},
  year={2022}
}

**Paper Topic And Main Contributions:**

Considering that the capability of pre-trained dialogue models in utilizing long-range contexts is constrained by the scarcity of long-turn dialogue sessions in the pre-trained dialogue corpus, this paper aims to automatically build a large-scale long-turn dialogue corpus by reorganizing short-turn dialogue sessions from existing dialogue corpus.

The main contributions made by the authors are: 1) they proposed the Retrieve, Reorganize and Rescale framework (i.e. Re^3Dial) to build a large-scale long-turn dialogue corpus by using existing short-turn dialogue sessions; 2) they proposed to train an Unsupervised Dense Session Retriever (UDSR) with contrastive learning to capture the global semantic relevance; 3) they experimented their proposed approach on three multi-turn Chinese dialogue benchmarks.

**Questions For The Authors:**

Question A. Would you be able to provide the results in terms of Distinct-1 in Table 2?

Question B. Why didn't the authors provide the results in terms of PPL, ROUGE-L and Distinct-2 in Table 3?

**Reasons To Accept:**

(1) The authors proposed to automatically construct large-scale long-turn dialogue corpus by retrieving, reorganizing and rescaling the existing short-turn dialogue sessions, which barely requires human annotations.

(2) In comparison to BM25 and Contriever, the proposed Unsupervised Dense Session Retriever (UDSR) with contrastive learning effectively discriminates the positive retrieved session from the incoherent negative session.

**Reasons To Reject:**

(1) The proposed approach was solely evaluated on three Chinese dialogue datasets, which raises concern about cherry-picking. It could be better if the authors would experiment with English dialogue datasets to further demonstrate its effectiveness.

(2) The experimental results are not satisfactory. As we can see from Table 2, the performance of Re^3Dial is worse than Original in terms of Distinct-2. In addition, the performance improvements of Re^3Dial over Original in terms of BLEU-1, BLEU-2, ROUGE-L and Distinct-2 are mostly marginal. Moreover, the authors did not provide the results in terms of Distinct-1, raising concern about cheery-picking.

(3) The performance of the proposed dense retriever is not promising. First of all ,the authors did not report PPL, ROUGE-L and Distinct-2 in Table 2, which is confusing. Moreover, as we can see in Table 2, improvements made by the proposed Re^3Dial over Random, BM25 and Contriever are somewhat marginal in terms of BLEU-1 and BLEU-2 especially under the setting 'Further Pr-training on LM' and 'Further Pre-training on DM'. More importantly, considering that the focus of this work is to build long-turn dialogue sessions by retrieving relevant short-turn dialogue sessions, it is very critical to train a retriever that really performs well. However, the authors only compared their retriever with two baseline retrievers including BM25 and Contriever, which fails to demonstrate the superiority of the proposed retriever.

**Reproducibility:**

4: Could mostly reproduce the results, but there may be some variation because of sample variance or minor variations in their interpretation of the protocol or method.

**Reviewer Confidence:**

3: Pretty sure, but there's a chance I missed something. Although I have a good feel for this area in general, I did not carefully check the paper's details, e.g., the math, experimental design, or novelty.

**Typos Grammar Style And Presentation Improvements:**

(1) In Table 11 on page 14, there is a typo in the first turn uttered by person A in Example #1. Please replace 'only make things words · · ·' with 'only make things worse · · ·'

(2) In Table 11 on page 14, there is another typo in the last turn uttered by person B in Example #1. Please replace 'girls should not be to humble in a relationship' with 'girls should not be too humble in a relationship,'

(3) In Figure 7 on page 15, there is a typo in the second paragraph in the Labeling Instructions. Please replace 'In the process of evaluation, you will be given on dialogue context' with 'In the process of evaluation, you will be given a dialogue context' or 'In the process of evaluation, you will be given one dialogue context'.

(4) In line 404 on page 6, there is a grammatical error in the sentence 'We compare different approaches to retrieve dialogue session'. Please replace it with 'We compare different approaches to retrieving dialogue session'.

---

> ### Author Rebuttal · Authors · 2023-08-28
>
> > 1. The proposed approach was solely evaluated on three Chinese dialogue datasets, which raises concern about cherry-picking. It could be better if the authors would experiment with English dialogue datasets to further demonstrate its effectiveness.
>
> Our framework is language-agnostic. To further verify the effectiveness of Re^3Dial in different languages, we conduct experiments on an English corpus. Specifically, we collect an English dialogue pre-training corpus, which consists of 1 million conversations from Reddit. The average number of turns in the original corpus is 2.3.
>
> We conduct further pre-training on a GPT-2 large model. We report PPL$_{zero-shot}$ on three widely-adopted English multi-turn dialogue benchmarks, including Blended Skill Talk [1], PersonaChat [2], and Wizard of Wikipedia [3]. From the results shown in the following table, we can see that Re$^3$Dial achieves significantly lower PPL than the Original baseline. These results demonstrate the effectiveness of Re$^3$Dial in English.
>
> | Pre-training Data | Blended Skill Talk | PersonaChat | Wizard of Wikipedia |
> | ----------------- | ------------------ | ----------- | ------------------- |
> | Original          | 33.72              | 39.61       | 50.05               |
> | Re$^3$Dial        | **29.17**              | **31.37**       | **48.62**               |
>
> > 2. As we can see from Table 2, the performance of Re^3Dial is worse than Original in terms of Distinct-2. Moreover, the authors did not provide the results in terms of Distinct-1, raising concern about cheery-picking.
>
> There is only one case where Re$^3$Dial does not outperform the original baseline (9.24 vs 9.27 in Distinct-2), which is also a comparable performance. We further present the average Distinct-1 metric in the table below.
>
> | Setting | Model | Dist-2 | Dist-1 |
> | ---- | ---- | ---- | ---- |
> | Pre-training From Scratch | Original | 7.17 | 0.90 |
> |                            | Re$^3$Dial | **7.88**  | **1.12** |
> | Further Pre-training on LM | Original   | 12.33     | **2.06**     |
> |                            | Re$^3$Dial | **12.59** | **2.06**     |
> | Further Pre-training on DM | Original   | 20.65     | 2.01     |
> |                            | Re$^3$Dial | **21.15** | **2.07** |
>
> > 3. The experimental results are not satisfactory. The performance improvements of Re$^3$Dial over Original in terms of BLEU-1, BLEU-2, ROUGE-L and Distinct-2 are mostly marginal.
>
> We would like to further clarify the substantial improvements of Re$^3$Dial.
>
> Firstly, for automatic evaluations (Section 4.3.1), Re$^3$Dial demonstrates substantial improvement in automatic metrics in an open-ended generation task (i.e., open-domain dialogue generation in our experiments). Let us reference the results in DuLeMon [4]. It was observed that increasing model parameters from 110M to 1.6B leads to an improvement of 0.0/0.6 in terms of BLEU-1/2, respectively.  In this light, the performance boost achieved by Re$^3$Dial is substantial. Furthermore, it is essential to underline that Re$^3$Dial outperforms the Original baseline in terms of PPL$_{zero-shot}$ and PPL by a large margin. As pointed out by [5][6][7], perplexity shows a stronger correlation with human judgments than other automatic metrics (e.g., BLEU).
>
> Secondly, considering that automatic metrics cannot fully demonstrate the performance of open-ended dialogue models [5][6][7][8], we further conduct human evaluations (Section 4.3.2). As illustrated in Figure 3, the Re$^3$Dial-trained dialogue model performs significantly better than the Original baseline in sensibleness and coherence in the long-turn test set.
>
>
> > 4.  The authors did not report PPL, ROUGE-L and Distinct-2 in Table 3, which is confusing.
>
> Taking further pre-training on DM for example, we present the detailed evaluation results of the dialogue model trained on the corpus constructed by different retrievers. We report the average metric over three benchmarks. As we can see, Re$^3$Dial achieves the best performance on all automatic metrics. These detailed evaluation results are omitted in the main paper for space considerations.
>
> | Retriever  | PPL$_{zero-shot}$ | PPL       | BLEU-1    | BLEU-2    | ROUGE-L   | Dist-1   | Dist-2    |
> | ---------- | ----------------- | --------- | --------- | --------- | --------- | -------- | --------- |
> | Original   | 59.60             | 22.80     | 18.82     | 9.98      | 18.17     | 2.01     | 20.65     |
> | Random     | 61.83             | 22.93     | 19.03     | 10.08     | 18.24     | 2.05     | 21.03     |
> | BM25       | 73.43             | 22.86     | **19.48**     | 10.32     | 18.28     | 1.99     | 20.74     |
> | Contriever | 61.48             | 22.98     | 19.34     | 10.22     | 18.06     | 1.97     | 20.49     |
> | Re$^3$Dial | **54.54**         | **22.06** | **19.48** | **10.37** | **18.58** | **2.07** | **21.15** |
>
> > 5. The performance of the proposed dense retriever is not promising. Moreover, as we can see in Table 3, improvements made by the proposed Re^3Dial over Random, BM25 and Contriever are somewhat marginal in terms of BLEU-1 and BLEU-2 especially under the setting 'Further Pre-training on LM' and 'Further Pre-training on DM'.
>
> We would like to further clarify the effectiveness of the dense retriever in Re$^3$Dial.
>
> Firstly, in the zero-shot setting, where the pre-trained dialogue model is directly deployed for applications, other retrievers inversely increase PPL in most cases (Table 3). In contrast, as shown in Table 3 and Figure 4, the Re$^3$Dial-trained model consistently achieves the lowest PPL, especially as the context length grows. These results indicate that Re$^3$Dial significantly outperforms other retrievers in enhancing the models' utilization of long-range contexts.
>
> Secondly, in the fine-tuning setting, despite that fine-tuning on sizable long-turn dialogues weaks the influence of pre-training [9], Re$^3$Dial still consistently outperforms all other baselines (Table 3). Notably, Re$^3$Dial achieves a substantially superior PPL than other retriever baselines (see the table above in Response4). And perplexity has shown a stronger correlation with human judgments than other automatic metrics (e.g., BLEU) [5][6][7]. Moreover, considering that the automatic metrics cannot fully demonstrate the performance of open-ended dialogue models [5][6][7][8], we further conduct a pair-wise human evaluation to study the model's performance trained with different retrievers. We conduct human evaluations on the long-turn test set (Section 4.3.2). The human evaluation results shown in the following table indicate that the dense retriever in Re$^3$Dial leads to significantly better performance.
>
> | Retriever                 | Sensibleness (Win) | Sensibleness (Lose) | Informativeness (Win) | Informativeness (Lose) |
> | ------------------------- | ------------------ | ------------------- | --------------------- | ---------------------- |
> | Re$^3$Dial vs. BM25       | 39                 | 18                  | 42                    | 20                     |
> | Re$^3$Dial vs. Contriever | 36                 | 20                  | 40                    | 21                     |
>
> Lastly, we also evaluate the retriever using individual tests in different aspects to gain a deeper understanding of the effectiveness of different retrievers. From the results shown in Table 4, we can see that our proposed dense retriever outperforms baselines by a large margin in capturing global relevance and discourse coherence.
>
> > 6.  More importantly, considering that the focus of this work is to build long-turn dialogue sessions by retrieving relevant short-turn dialogue sessions, it is very critical to train a retriever that really performs well. However, the authors only compared their retriever with two baseline retrievers including BM25 and Contriever, which fails to demonstrate the superiority of the proposed retriever.
>
> We select BM25 and Contriever as the baselines since they are the previous state-of-the-art non-parametric retriever and dense retriever [10].
>
> We additionally conduct experiments based on an encoder for sentence similarity (Sentence-BERT) [11].
> We report the average PPL$_{zero-shot}$ over three benchmarks. As we can see, Sentence-BERT underperforms Contriever and also significantly underperforms the dense retriever in Re$^3$Dial.
> | Retriever  | Pre-training From Scratch | Further Pre-training on LM | Further Pre-training on DM |
> | ---------- | ------------------------- | -------------------------- | -------------------------- |
> | Original   | 193.51                    | 13.06                      | 59.60                      |
> | Random     | 170.58                    | 14.92                      | 61.83                      |
> | BM25       | 192.94                    | 14.69                      | 73.43                      |
> | Sentence-BERT   | 161.05                    | 13.76                      | 63.39                      |
> | Contriever | 154.65                    | 13.41                      | 61.48                      |
> | Re$^3$Dial | **135.88**                    | **12.25**                      | **54.54**                      |
>
> >7. Typo and missing references.
>
> Thanks for pointing these out. We will fix them in our revised version of the paper.
>
>
> [1] Can You Put it All Together: Evaluating Conversational Agents’ Ability to Blend Skills. ACL2020
>
> [2] Personalizing Dialogue Agents: I have a dog, do you have pets too? ACL2018
>
> [3] Wizard of Wikipedia: Knowledge-Powered Conversational agents. ICLR2019
>
> [4] Long Time No See! Open-Domain Conversation with Long-Term Persona Memory. ACL2022 findings.
>
> [5] Towards a Human-like Open-Domain Chatbot. arXiv2020
>
> [6] How not to evaluate your dialogue system: An empirical study of unsupervised evaluation metrics for
> dialogue response generation. EMNLP2016
>
> [7] Towards an Automatic Turing Test: Learning to Evaluate Dialogue Responses. ACL2017
>
> [8] RUBER: An Unsupervised Method for Automatic Evaluation of Open-Domain Dialog Systems. AAAI2018
>
> [9] Long Text Generation by Modeling Sentence-Level and Discourse-Level Coherence. ICLR2019
>
> [10] Unsupervised Dense Information Retrieval with Contrastive Learning. TMLR2022
>
> [11] Sentence-BERT: Sentence Embeddings using Siamese BERT-Networks. EMNLP2019

---

### Meta-Review · Area_Chair_1HXU · 2023-09-08

**Recommendation:** 5

**Metareview:**

This paper addresses the data scarcity problem of long-turn dialogues in pretraining large-scale open-domain dialogue models. The authors introduce a novel and effective "Retrieve, Reorganize, and Rescale" framework that can automatically construct billion-scale long-turn dialogues by reorganizing existing short-turn dialogues. The proposed method has been evaluated on Chinese dialogue datasets, effectively demonstrating its efficacy. The authors have committed to releasing their model, toolkit, and data for public use.

The soundness scores for this paper are as follows: (4, 4, 4, 4, 3). Initially, reviewers expressed concerns that evaluating the method using only a single language might not validate the generalizability of the approach. However, in the rebuttal, the authors presented additional experiments on an English benchmark, which addressed these concerns, leading all reviewers to agree on the robustness of the work.

The excitement scores for this paper are: (3, 3, 4, 4, 4). The majority of reviewers show enthusiasm for this work, primarily because the proposed data augmentation method can be adapted to various languages without requiring manual intervention. This indicates the significant potential and broad applicability of the work.

---

### Decision · Program_Chairs · 2023-10-07

**Decision:**

Accept-Main

**Comment:**

This paper addresses the data scarcity problem of long-turn dialogues in pretraining large-scale open-domain dialogue models. The authors introduce a novel and effective "Retrieve, Reorganize, and Rescale" framework that can automatically construct billion-scale long-turn dialogues by reorganizing existing short-turn dialogues. The proposed method has been evaluated on Chinese dialogue datasets, effectively demonstrating its efficacy. The authors have committed to releasing their model, toolkit, and data for public use.

The soundness scores for this paper are as follows: (4, 4, 4, 4, 3). Initially, reviewers expressed concerns that evaluating the method using only a single language might not validate the generalizability of the approach. However, in the rebuttal, the authors presented additional experiments on an English benchmark, which addressed these concerns, leading all reviewers to agree on the robustness of the work.

The excitement scores for this paper are: (3, 3, 4, 4, 4). The majority of reviewers show enthusiasm for this work, primarily because the proposed data augmentation method can be adapted to various languages without requiring manual intervention. This indicates the significant potential and broad applicability of the work.